# Composition of Gut Microbiota in Children with Autism Spectrum Disorder: A Systematic Review and Meta-Analysis

**DOI:** 10.3390/nu12030792

**Published:** 2020-03-17

**Authors:** Lucía Iglesias-Vázquez, Georgette Van Ginkel Riba, Victoria Arija, Josefa Canals

**Affiliations:** 1Department of Preventive Medicine and Public Health, Faculty of Medicine and Health Science, Universitat Rovira i Virgili, 43201 Reus, Spain; lucia.iglesias@urv.cat (L.I.-V.); victoria.arija@urv.cat (V.A.); 2Department of Psychology, Faculty of Education Sciences and Psychology, Universitat Rovira i Virgili, 43007 Tarragona, Spain; georgette.vanginkel@urv.cat

**Keywords:** gut microbiota, dysbiosis, autism spectrum disorder, ASD, children, adolescents, systematic review and meta-analysis

## Abstract

Background: Autism spectrum disorder (ASD) is a public health problem and has a prevalence of 0.6%–1.7% in children. As well as psychiatric symptoms, dysbiosis and gastrointestinal comorbidities are also frequently reported. The gut–brain microbiota axis suggests that there is a form of communication between microbiota and the brain underlying some neurological disabilities. The aim of this study is to describe and compare the composition of gut microbiota in children with and without ASD. Methods: Electronic databases were searched as far as February 2020. Meta-analyses were performed using RevMan5.3 to estimate the overall relative abundance of gut bacteria belonging to 8 phyla and 17 genera in children with ASD and controls. Results: We included 18 studies assessing a total of 493 ASD children and 404 controls. The microbiota was mainly composed of the phyla Bacteroidetes, Firmicutes, and Actinobacteria, all of which were more abundant in the ASD children than in the controls. Children with ASD showed a significantly higher abundance of the genera *Bacteroides*, *Parabacteroides*, *Clostridium*, *Faecalibacterium,* and *Phascolarctobacterium* and a lower percentage of *Coprococcus* and *Bifidobacterium*. Discussion: This meta-analysis suggests that there is a dysbiosis in ASD children which may influence the development and severity of ASD symptomatology. Further studies are required in order to obtain stronger evidence of the effectiveness of pre- or probiotics in reducing autistic behaviors.

## 1. Introduction

Autism spectrum disorder (ASD) is a chronic neurodevelopmental disorder of early onset and heterogeneous etiology [1]. The prevalence of ASD has been steadily increasing in recent years, which may to some extent be due to greater awareness of the disease on the part of health and education professionals, the increased availability of diagnoses, and changes in diagnostic criteria. However, the recently identified interaction between some environmental factors and ASD also helps us to understand this increase in prevalence [2,3]. Estimates report a prevalence of ASD of between 0.6% and 1.7% in children and adolescents, representing a serious public health problem. It has also been observed that males are up to 4 times more likely than females to be diagnosed with ASD [4,5,6].

According to the Diagnostic and Statistical Manual of Mental Disorders (5th ed.; DSM-5) [7], affected individuals have persistent deficiencies in social communication and interaction and are characterized by restrictive and repetitive behavior patterns, interests, or activities. In addition to these main symptoms, people with ASD tend to suffer from comorbidities such as intellectual disability, gastrointestinal (GI) problems, and eating and sleeping disorders [8,9,10]. Regarding the GI problems (i.e., constipation, abdominal pain, diarrhea, gas, and vomiting), prevalence of these ranges from 9% to 90% in people with ASD, which is a much higher rate than in neurotypical individuals [9]. Some studies have even observed an association between the GI symptoms and the severity of the clinical manifestations of the ASD, which means that the autistic symptomatology would be more frequent and severe in children with comorbid GI problems than in those without [9,11,12]. As far as the concept of the “gut–brain microbiota axis” is concerned, current scientific advances postulate that the gut microbiota plays a role in brain development and function through the endocrine, immune, and nervous systems [13]. Therefore alterations in the gut microbiota could trigger not only some of the GI symptoms that autistic children suffer from but also some of their neuropsychiatric symptoms [13,14]. In an attempt to clarify the role of gut microbiota in the appearance and development of ASD, some clinical studies have observed that autistic subjects, as opposed to neurotypical subjects, suffer from dysbiosis regarding both the type and abundance of gut bacteria [15,16,17,18,19]. No differences in the bacterial profile were found in other case–control studies including ASD and neurotypical siblings [20,21]. Despite its controversial results, a previous meta-analysis [22] suggested there was an association between ASD and alterations in microbiota composition, thereby highlighting the need for additional cohort studies aimed at evaluating this association. On the basis of these findings, a more thorough evaluation of the intestinal microbiota would be expected to help individualize microbiological interventions, and this could serve as a complementary treatment for ASD. Indeed, some clinical trials and animal studies have reported changes in neurological function, behavior, and comorbid symptoms of autistic children after rebalancing the composition of the gut microbiota through the use of antibiotics, prebiotics, and probiotics or the transplanting of fecal microbiota [22,23,24,25].

Given that the previous meta-analysis [22] was out-of-date at the time of publication and included a limited number of studies, the aim of the present systematic review and meta-analysis is to update current findings about the composition of gut microbiota in children and adolescents with ASD. The differences between the gut microbiota of children with ASD and their neurotypical counterparts are also studied. In view of the emergence of new therapies based on the modulation of gut microbiota, characterizing the individual gut bacterial profile could help improve nutritional interventions and provide a better quality of life for subjects with ASD.

This study is registered on PROSPERO (https://www.crd.york.ac.uk/prospero/) under ID number CRD42018093461.

## 2. Material and Methods

### 2.1. Literature Search and Selection Criteria

This systematic review was carried out following the Meta–analysis of Observational Studies in Epidemiology (MOOSE) guidelines [26]. Two authors (LIV and GVG) performed independent searches in PubMed, Scopus, and Cochrane Library electronic databases as far as February 2020, looking for studies that reported the composition of gut microbiota of children and adolescents with and without ASD. The search strategy was as follows: ((“Child Development Disorders, Pervasive” [Mesh] OR “Autism Spectrum Disorder” [Mesh] OR ASD OR PDD OR Autism OR “Autistic Disorder” OR “Autism Spectrum Disorder” OR “Asperger Syndrome” OR “Asperger Disorder” OR “Autistic traits”) AND (Microbiota [Mesh] OR Microbiome OR Microflora OR Dysbiosis OR “Fecal microbiota” OR Gastrointestinal OR “Intestinal flora”)). There were no restrictions in terms of language or year of publication, but studies carried out on adults and animals were excluded. The reference lists of original studies and reviews were hand-searched for additional studies of interest.

The titles and abstracts of the articles thrown up by the search were assessed and the full texts of those that were potentially relevant were carefully read. Those that met the following inclusion criteria were collected for the meta-analysis: (a) observational studies and controlled trials, (b) reporting the microbiota composition of children and adolescents, (c) assessment of participants with ASD and controls, and (d) available data regarding the abundance of bacteria. Case reports, systematic reviews, meta-analyses, animal studies, and research that assessed adults were excluded from the selection process.

### 2.2. Data Extraction and Quality Assessment

The following data were extracted from the included studies by two authors (LIV and GVG) working independently: first author’s surname, year of publication, country, study design, sample size and age of children with ASD and controls, and data on microbiota (including the phyla and genera of bacteria detected and the methodology used for the microbiology assessment). We also looked at whether the authors assessed diet and the use of probiotics.

As the Cochrane Non-Randomized Studies Methods Group recommends, the methodological quality of the studies was assessed using the Strengthening the Reporting of Observational Studies in Epidemiology (STROBE) statement [27]. This scale categorizes the final score obtained by the studies into good, fair, and poor quality according to the following three domains: selection, comparability, and outcome. The quality of the present meta-analysis was also assessed, using the PRISMA guide [28].

### 2.3. Data Analyses

The included studies reported the percentage, also known as the relative abundance, of bacteria observed in children and adolescents with and without ASD. They also reported the mean, standard error (SE), or confidence interval for these measurements. We standardized all the extracted data and used the relative abundance and SE from each study to obtain the overall percentage of bacteria from different phyla and genera in children with ASD and in the controls. For each bacterial phyla and genera, the difference in bacterial percentage between the ASD and control groups was calculated in order to know the relative abundance of bacteria in children with ASD compared to those with neurotypical development.

A random-effect meta-analysis was undertaken using Review Manager 5.3 software (The Cochrane Collaboration, Copenhagen, Denmark), selecting the inverse-variance method. Heterogeneity was assessed by calculating I^2^ [29]. The use of forest plots allowed us to visualize the results. Sensitivity analyses were also conducted by removing one study at a time in order to assess the robustness of the results. Statistical significance was set at *P* < 0.05 for all the analyses.

## 3. Results

The selection process for the included studies is shown in Figure 1. The search strategy identified 706 publications that were first scrutinized by title and abstract. Of these, 78 were considered to be potentially relevant. The full texts of 21 articles were read after 57 studies had been excluded for the following reasons: animal studies, not assessing the association under study, reviews and data not available. In the end, 18 articles reporting on microbiota in children and adolescents with and without ASD were included in the meta-analysis. Authors were contacted when the required data were not available. In this regard, although the study by Sun et al. [30] met the main inclusion criteria, it was not included in the meta-analysis due to the lack of necessary data and no response from the authors.

### 3.1. Study Characteristics

Table 1 summarizes the characteristics of the studies included, which were conducted between 2010 and 2019. The greatest numbers were performed in the United States [11,12,21,31,32,33], Australia [20,34,35], Italy [17,36,37], and China [16,38,39], with one study each from Japan [40], India [18], and Spain [41]. Sample size ranged from 6 to 58 and the participants’ ages from 2 to 18 years. In total, this review pooled results from 493 subjects with ASD and 404 age-matched neurotypical controls. Most of the investigations were observational, including five cohort studies; one was a randomized controlled trial. Regarding the microbiota assessment, most classified the bacteria detected according to both phylum and genus, with a wide variety of bacteria being studied. Few studies included the level of species when classifying the bacteria detected, so it was not possible to group the data at that level in the meta-analysis. Two of the earliest studies in this field used cultures [11,20] to evaluate bacterial diversity. In subsequent studies, however, authors opted for genetic techniques such as different types of polymerase chain reaction (PCR) [12,16,17,18,21,34,35,38,39,41]. The technique of pyrosequencing was also used by a number of authors [31,32,33,36,37,40]. We can see in Table 1 that about half of the researchers took into account diet and the use of probiotics when evaluating the microbiota. As to the methodological quality, according to the STROBE checklist, good quality was reported in 15 studies [11,12,16,17,18,20,21,32,33,36,37,38,39,40,41] and fair quality in the other 3 [31,34,35].

The results of the meta-analysis including the two levels of bacterial classification (phylum and genus) are presented in Table 2. The phyla are shown in bold and the genera in italics, grouped under the phyla to which they belong. The forest plots of relevant meta-analyses at genus level are also shown as figures. The forest plots of bacterial phyla and those at genus level that were not statistically significant are provided as Appendix A.

### 3.2. Bacterial Phyla More Abundant in ASD Children

#### 3.2.1. Bacteroidetes

In the meta-analysis of Bacteroidetes we included 12 studies, arriving at the following results: 14.33% in children with ASD (95% CI: 12.79, 15.87) and 10.97% in the control group (95% CI: 9.57, 12.36). Between-study heterogeneity was 100% in both the ASD and the control group and was also very high (90%) when the subgroups were compared. The effect size was significant and large (Z = 25.57, *P* = 0.002).

#### 3.2.2. Firmicutes

Eleven studies were included in the meta-analysis of Firmicutes, leading to these results: 13.42% in children with ASD (95% CI: 12.50, 14.34) and 10.77% in the control group (95% CI: 9.89, 11.64). The heterogeneity between studies was 100% in ASD children and the control group and 94% when the two groups were compared. The effect size was significant and large (Z = 25.82, *P* < 0.001). The ratio of Bacteroidetes to Firmicutes was higher in ASD children (0.69) than in controls (0.44).

#### 3.2.3. Proteobacteria

The relative abundance of Proteobacteria was also assessed on the basis of 11 trials. The percentage in children with ASD was 0.09% (95% CI: 0.05, 0.13) and 0.02% in the control group (95% CI: 0.00, 0.03). Between-study heterogeneity was very high (97% and 96% respectively) and also very high between the subgroups (I^2^ = 92.6%). The effect size was moderate and significant (Z = 4.20, *P* < 0.001).

#### 3.2.4. Tenericutes

Seven studies were included in the meta-analysis of Tenericutes. The forest plot (Appendix A) showed the following results: just 0.06% (95% CI: 0.04, 0.07) of detected microbiota in children with ASD, with no evidence of the presence of bacteria belonging to this phylum being observed in children from the control group. Between-study heterogeneity was very high (I^2^ = 99%) in the ASD group and relatively low (I^2^ = 28%) in the controls. It was also therefore high when the two groups were compared (I^2^ = 97.7%). The effect size was large and significant (Z = 5.38, *P* < 0.001).

### 3.3. Bacterial Phyla with No Difference between ASD and Controls

The meta-analysis for Actinobacteria, Cyanobacteria, Fusobacteria, and Verrucomicrobia showed non-significant differences between children with ASD and the controls, and the relative abundance of most of these as part of the total detected microbiota was negligible (Appendix A). Between-study heterogeneity was high, ranging between 70% and 100% in all the analyses, although it became null in all cases when the ASD and control groups were compared. The I^2^ also became null between the subgroups when they were compared in the analyses of both phyla. The effect sizes ranged from 2.66 in Cyanobacteria to 7.86 in Actinobacteria and were always statistically significant. Despite the fact that the evidence in these cases was not enough for a statistically significant difference between the groups to be observed, the difference in bacterial percentage between the ASD and the controls indicates that children with ASD had a greater abundance of Actinobacteria, Cyanobacteria, and Fusobacteria and a lower abundance of Verrucomicrobia.

No significant differences were observed regarding heterogeneity or the result of the meta-analyses following the sensitivity analyses of any phyla.

### 3.4. Bacterial Genera More Abundant in ASD Children

#### 3.4.1. Bacteroides

From the 12 trials included in this meta-analysis (Figure 2) we obtained that the level of *Bacteroides* was 9.04% in children with ASD (95% CI: 7.62, 10.47) and 4.69% in the control group (95% CI: 3.67, 5.71). There was very high heterogeneity (I^2^ = 99% in both groups) between the included studies and also between the subgroups (I^2^ = 95.8%). Nevertheless, the overall effect size was large and significant (Z = 15.56, *P* < 0.001). The difference in bacterial percentage between the ASD and control groups was 1.93, indicating that the percentage of *Bacteroides* was much higher in children with ASD compared to the controls.

#### 3.4.2. Parabacteroides

The random-effects meta-analysis of *Parabacteroides* involved 10 studies (Figure 3) and resulted in the following percentages: 0.32% (95% CI: 0.21, 0.43) in children with ASD and 0.04% (95% CI: 0.08, 0.15) in the control group. Between-study heterogeneity was high in both subgroups (I^2^ = 95% and 92% respectively) and also when we compared the two groups (I^2^ = 95.3%). The overall effect size was large and highly significant (Z = 5.69, *P* < 0.001). The difference in bacterial percentage between the ASD and control children was 8, clearly indicating that the percentage of *Parabacteroides* was many times higher among the former.

#### 3.4.3. Faecalibacterium

Twelve studies were included in the meta-analysis of *Faecalibacterium* (Figure 4). It accounted for 6.84% of the total detected microbiota in children with ASD (95% CI: 5.34, 8.35) and 5% in the control group (95% CI: 4.15, 5.85). Heterogeneity was 99% between the studies for both ASD and the control group and was also high (I^2^ = 77%) between the two groups. The overall effect size was large and significant (Z = 16.22, *P* < 0.001). The difference in bacterial percentage between the ASD and control groups was 1.37, showing that *Faecalibacterium* was more abundant among the ASD children than in the controls.

#### 3.4.4. Clostridium

The random-effects meta-analysis of *Clostridium* included 10 studies (Figure 5) and showed that the percentages were 0.74% (95% CI: 0.44, 1.05) in children with ASD and 0.16% (95% CI: 0.06, 0.26) in the control group. Heterogeneity was high between studies (I^2^ = 97% in ASD and 98% in controls) and also between subgroups (I^2^ = 92.2%). The overall effect size was large and highly significant (Z = 5.87, *P* < 0.001). The difference in bacterial percentage was 4.63, which indicates that *Clostridium* was much more abundant in ASD children than in the controls.

#### 3.4.5. Phascolarctobacterium

Six trials were included in the random-effects meta-analysis of *Phascolarctobacterium* (Figure 6). The results showed percentages of 0.13% (95% CI: 0.03, 0.24) in children with ASD and 0.01% (95% CI: 0.00, 0.02) in the control group. Heterogeneity between studies was high in both the ASD and control groups (I^2^ = 93% and 89% respectively) and also between the two groups (I^2^ = 80.3%). The overall effect size was moderate and significant (Z = 3.42, *P* = 0.001). The difference in bacterial percentage between the ASD and control children was 13, indicating that the genus *Phascolarctobacterium* was much more abundant in ASD children than in the neurotypical participants. Sensitivity analyses revealed that the heterogeneity fell from 93% to 64% in the ASD group when the study by Kang et al. from 2017 (a) was removed from the meta-analysis, although this brought no change in the control group. The difference between the two groups was small and became non-significant when the meta-analysis did not include the research by Kang et al.

### 3.5. Bacterial Genera Less Abundant in ASD Children

#### 3.5.1. Coprococcus

Eight trials were included in the random-effects meta-analysis of *Coprococcus* (Figure 7). The results showed the levels to be 0.11% (95% CI: 0.07, 0.15) in children with ASD and 0.24% (95% CI: 0.16, 0.32) in the control group. Between-study heterogeneity was high in both subgroups (I^2^ = 97% and 98% respectively) and also when the two groups were compared (I^2^ = 88.1%). The overall effect size was large and highly significant (Z = 8.43, *P* < 0.001). The difference in bacterial percentage was 0.46, indicating that the level of *Coprococcus* was lower among children with ASD compared to the neurotypical individuals.

#### 3.5.2. Bifidobacterium

We included 12 trials in the meta-analysis of *Bifidobacterium* (Figure 8), which indicated that the levels of this genus as part of the total detected microbiota was 0.46% in children with ASD (95% CI: 0.33, 0.59) and 0.89% in the control group (95% CI: 0.72, 1.05). Heterogeneity was very high between the studies included (I^2^ = 99% in both groups) and also between the subgroups (I^2^ = 93.9%). The overall effect size was large and significant (Z = 13.08, *P* < 0.001). The difference in bacterial percentage between the ASD and control groups was 0.52, which means that *Bifidobacterium* was less abundant in children with ASD than in the controls.

### 3.6. Bacterial Genera with No Difference between ASD and Controls

The available evidence was not enough to show a statistically significant difference between the ASD and control groups regarding some bacterial genera. Nevertheless, the abundance of *Anaerotruncus*, *Ruminococcus*, *Dorea,* and *Roseburia* were higher in ASD children than in the control group. Of these, *Ruminococcus* was the most abundant and most studied genus, with 11 trials being included in its meta-analysis resulting in the following overall relative abundances: 2.90% (95% CI: 2.22, 3.58) in participants with ASD and 2.21% (95% CI: 1.50, 2.92) in the control group. Heterogeneity was high between the subgroups (96% and 98% respectively) and moderate when they were compared (I^2^ = 47.3%) for *Ruminococcus*. The other three genera were less present in the microbiota studied, with an overall relative abundance of under 0.5% in both the ASD and control groups. Between-study heterogeneity fell within the range 90%–100% in all cases.

The overall percentage of *Anaerostipes*, *Blautia*, *Veillonella*, *Dialister*, *Sutterella,* and *Akkermansia* was lower in the ASD than in the control group. The studies reported that *Sutterella* had the highest relative abundance but only in the control group (2.21%), whereas among the ASD children it was very low (0.11%). Despite this apparently large difference, it was not statistically significant. The next most abundant was the genus *Blautia*, whose meta-analysis provided the following results: 1.52% in the ASD group (95% CI: 0.86, 2.19) and 1.91% in the control group (95% CI: 0.61, 3.21). Percentages were very low for all the others, ranging from near zero to 0.65%. Heterogeneity was moderate to high in all cases in both the ASD and control groups.

## 4. Discussion

The present study updates the last published meta-analysis [22], including a greater number of studies and providing a more comprehensive and stronger overview of current knowledge about the composition of the gut microbiota in children with and without ASD and the differences between them. We conducted a systematic review of the available literature on the subject in order to collect data as up-to-date as possible. In addition, we used a meta-analysis to pool the results of 18 studies with fair and high-quality ratings and to provide joint information on the relative abundance of bacteria belonging to 8 phyla and 17 genera in almost 500 children with ASD and over 400 controls. Most of the reviewed studies had a low to medium sample size, which is a valuable strength given the difficulties involved in conducting epidemiological studies with children, especially when they have some kind of disability. Nevertheless, some limitations continued through to the meta-analysis and should be considered. First, the study design, methodological quality, and age and gender of the children contributed to increasing the heterogeneity of the meta-analyses. Given that the environment along with dietary and cultural habits influences microbiota composition, the inclusion of studies from all over the world could be the reason why the between-study heterogeneity values we obtained were so high. Another limitation was the impossibility of evaluating bacterial diversity at the species level; few studies were found that gave this information, and those that did reported data in relation to different species. Moreover, all of the included studies assessed the abundance of bacteria in stool samples, which may underestimate bacterial diversity because only those bacteria that are shed from the intestinal wall can be evaluated. Although it would make it more difficult to obtain samples, greater diversity has been observed in microbiota isolated from biopsies than from feces. The bacterial profile of the gut microbiota is unique to each individual and also varies in the same person according to age, lifestyle, and dietary habits [42]. There is, therefore, no optimal composition of the gut microbiota. However, balance and diversity within the bacterial population are crucial for proper physiological functioning, especially as regards the immune, metabolic, and nervous systems. Imbalances in gut bacterial composition (e.g., through a marked decrease in beneficial bacteria) allow potentially harmful bacteria to colonize the intestinal tract to the extent that dysbiosis has been described as being linked to several gastrointestinal problems, as would be expected, but can also trigger extra-intestinal physiological problems.

According to our findings, the microbiota of the children assessed in the included studies was mainly composed of the phyla Bacteroidetes and Firmicutes, followed by Actinobacteria. The relative abundance of Proteobacteria, Verrucomicrobia, Cyanobacteria, Fusobacteria, and Tenericutes was much lower and even close to zero in individuals with ASD and the controls. Returning to main phyla, although the overall results of the meta-analyses indicated that all were more abundant in ASD children than in neurotypical subjects, controversial results have been observed between the studies. For Bacteroidetes, while most of them found a higher relative abundance in children with ASD [16,17,18,31,32,39], some other reported opposite results [20,37,41]. The same occurred for Firmicutes, with some studies reporting higher percentages in ASD [20,37,41], others in controls [16,31,39], and others suggesting even no differences between the groups [17,21,32]. In the case of Actinobacteria, the overall relative abundance was not significantly different between ASD and controls. At genus level and compared with neurotypically developed children, those with ASD showed a significantly greater abundance of *Bacteroides*, *Parabacteroides*, *Clostridium*, *Faecalibacterium,* and *Phascolarctobacterium*. *Coprococcus* and *Bifidobacterium,* on the other hand, were significantly more abundant in the controls than in the ASD children. However, it must be said that, similar to the phyla, opposing findings were observed between the studies included in the meta-analysis for all genera, to a greater or lesser extent. The evidence was not enough for us to observe a statistically significant difference for *Akkermansia*, *Sutterella*, *Anaerostipes*, *Dialister*, *Blautia,* and *Veillonella*, although their percentages were higher in neurotypical children. Similarly, despite *Anaerotruncus*, *Ruminococcus*, *Dorea,* and *Roseburia* are more abundant in the ASD children than in the controls, it cannot be said that the difference was statistically significant.

Based on our analyses, therefore, and in line with previous reports [22], children with ASD in comparison with neurotypical children show dysbiosis as regards certain bacterial groups. The alterations in gut microbiota were related not only to the comorbid GI problem but also to the intensity of the autistic symptomatology, suggesting that bacterial activity and the resulting metabolites may be involved in the development and severity of ASD. These findings are the basis of the “gut–brain microbiome axis” concept, which postulates that there is a bidirectional interplay between gut bacteria and the brain. Although it is known that this communication occurs by means of hormones and neurotransmitters released by the gut endocrine system depending on the microbiota and metabolites that exist, the specific underlying mechanisms of this physiological interaction are not entirely clear.

In view of the results obtained, the two main issues in connection with dysbiosis in ASD are the great abundance of harmful bacteria and the low presence of beneficial bacteria. It has been widely observed in previous studies that there is a higher abundance of *Clostridium* in children with ASD than in neurotypical children [15,16,35,37,43]. Spore-forming bacteria like *Clostridium* release pro-inflammatory toxins that can reach the brain through blood flow [44]. Similarly, some of the metabolites derived from the activity of *Clostridiales* have been associated with repetitive behaviors and GI problems in ASD, which can be reversed following antibiotic use [44]. Another interesting point involves short chain fatty acids (SCFAs), produced from fiber fermentation. SCFA are involved in the proper functioning of the gut immune system through the modulation of gene expression. Any imbalance in the concentration of SCFA can, therefore, alter gut homeostasis and trigger peripheral inflammation. They also reach the brain through blood flow, influencing its development by modulating serotonin and dopamine production [45]. In this regard propionic acid (PPA), mainly produced by *Bacteroidetes*, is one of the major neurotoxic SFCA when its concentration rises [46,47]. This is consistent with the findings of Finegold et al. [31], who observed high levels of PPA in ASD children who had a high abundance of *Bacteroides* and *Clostridium* compared to healthy children. Furthermore, the relationship between high concentrations of PPA and behavioral disorders has been confirmed in various studies on rodents [46,48,49]. Also related to immune system imbalance as a mechanism that is damaging for brain functioning, cytokines have been described as the common language between the immune system and the nervous system [50]. On this subject it has been observed that *Faecalibacterium* is related to the up- and down-regulation of some of the genes involved in the expression of interferon (IFN) gamma [45], a cytokine to which exposure during fetal development has been related to ASD [51]. As an underlying mechanism, it is known that IFN-gamma plays an indirect role in brain plasticity and synapse formation [52]. Changes in microbiota may ultimately alter brain signaling, inducing some ASD behaviors by changing IFN-gamma concentrations [50].

In combination with the increased abundance of harmful bacteria, specifically the pro-inflammatory genus *Clostridium*, some studies have reported a lesser abundance of protective bacteria such as *Bifidobacterium* in ASD children [17,35,41,53]. It is known that some species of *Bifidobacterium* produce GABA [45], concentrations of which have therefore been found to be low in ASD children. GABA are closely related to the glutamate metabolism, which is the main excitatory neurotransmitter in the brain [54]. According to some studies, lower glutamate concentrations correlate with the severity of the anxiety and social and behavioral disorders typical of ASD [55,56]. Based on this, it has been suggested that GABA/glutamate abnormalities could play an important role in ASD pathology [55,57]. Amino acid dysregulation has also been reported in autistic children [58], and new evidence has highlighted the role of gut microbiota in amino acid metabolism [59]. Alterations to the metabolism of tryptophan (Trp) and serotonin has been found in ASD, with an increase in serotonin levels in more than 25% of ASD children [60,61]. Trp is the precursor of serotonin, which is a neurotransmitter involved in almost all behaviors including appetite, sleep, emotions, and cognitive and social skills [62]. Returning to the microbiota, animal studies have reported that the serotonergic systems in the brain are influenced by gut bacterial composition, which in turn leads to GI problems and emotional disorders [60,63]. It is therefore expected that when the abundance of Trp-metabolizing bacteria such as *Bacteroides* and *Clostridium* increases, serotonin concentrations also increase as a result of their metabolic activity [64,65]. Meanwhile, lower levels of glutathione, homocysteine, methionine, and S-adenosylmethionine have been observed more in ASD children than in neurotypical children [66,67]. Some of these are involved in the metabolism of sulfur and methylation, which contribute to the reduction of oxidative stress, cell detoxification, and excretion of heavy metals [45]. The amino acid dysregulation in ASD children could, therefore, be another mechanism underlying the development and severity of autistic symptomatology.

Furthermore, animal studies and clinical trials with probiotics containing strains of *Lactobacillus* and *Bifidobacterium* have both reported improvements in mood state, anxiety, sleep quality, and depression [68,69,70]. Evaluation of the gut bacterial profile could open a window to the possible treatment of autistic symptomatology and comorbidities through modulation of the gut microbiota by administering pre- or probiotics. This may increase the chances of individualized and successful therapies, increasing the health quality of people suffering from ASD. However, the results are still controversial, and further studies are required to assess the effect of what are known as “psychobiotics” on neurological disorders [25,71,72,73].

The present meta-analysis has reported a dysbiotic bacterial profile in ASD children. In short, when compared to neurotypical children, those with ASD showed a greater abundance of Bacteroidetes (*Bacteroides* and *Parabacteroides*) and some Firmicutes genera (specifically *Clostridium*, *Faecalibacterium* and *Phascolarctobacterium*) along with a lower abundance of *Coprococcus* and *Bifidobacteria*. According to the clinical findings, inflammation and dysfunction in the immune system mediated by microbiota composition are key elements in the development of GI problems and other extra-intestinal diseases such as ASD. However, the direction of causation, that is, whether the alterations in the microbiota lead to inflammation and imbalances in the immune system or vice versa, is still being studied. We have a growing knowledge of the interplay between gut microbiota, GI problems, and the physiopathology and symptomatology of ASD, but more research is needed before we can fully understand the physiological communication between the gut and brain. Future studies should look into several environmental factors that could affect the gut bacterial composition, including lifestyle, diet, exposure to environmental chemicals, and the use of antibiotics, pre- or probiotics.

## Figures and Tables

**Figure 1 nutrients-12-00792-f001:**
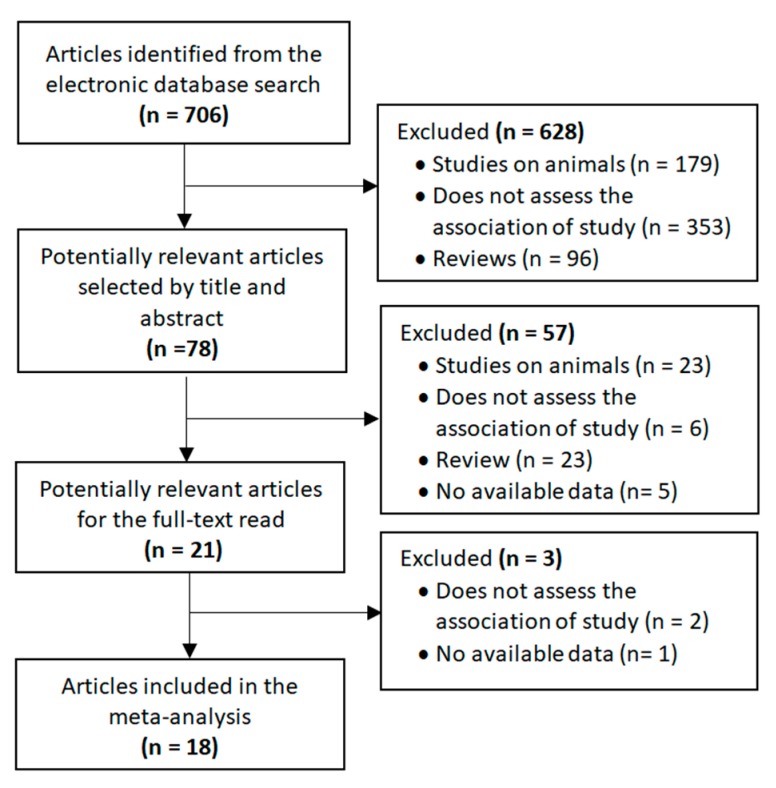
Flow diagram of selected studies.

**Figure 2 nutrients-12-00792-f002:**
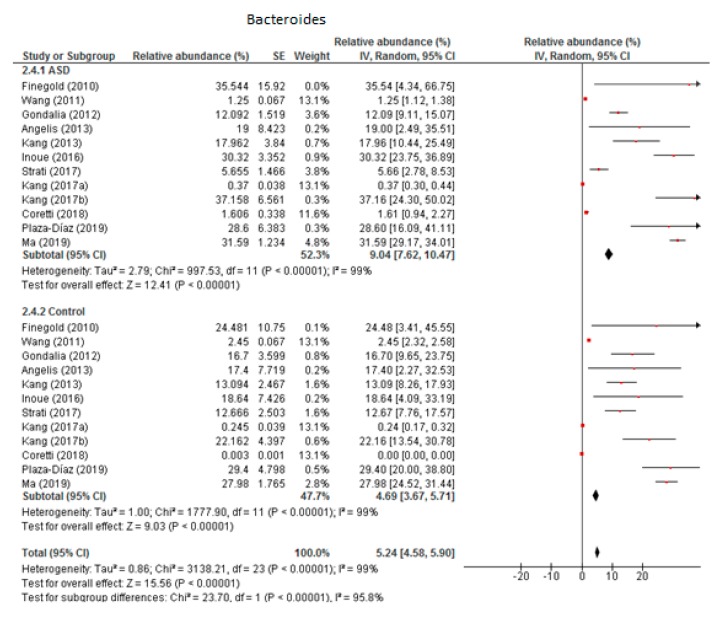
Forest plot of relative abundance of Bacteroides in children with ASD and controls.

**Figure 3 nutrients-12-00792-f003:**
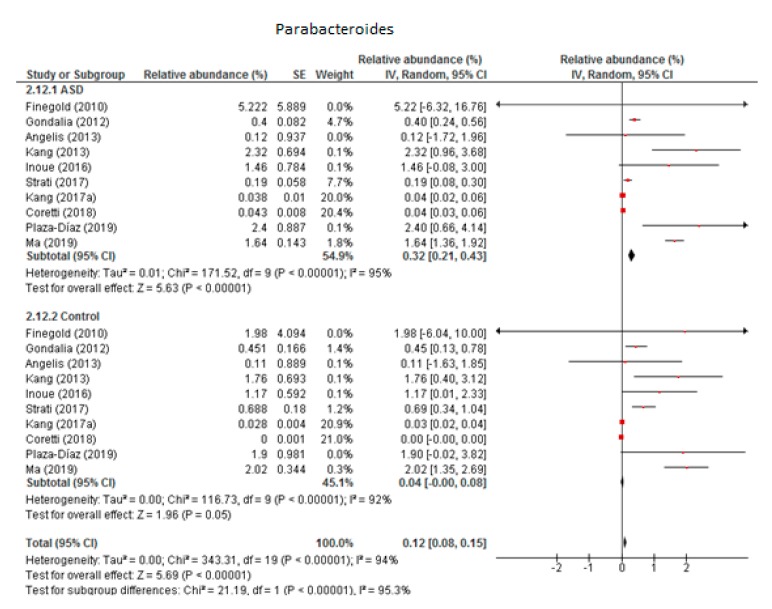
Forest plot of relative abundance of Parabacteroides in children with ASD and controls.

**Figure 4 nutrients-12-00792-f004:**
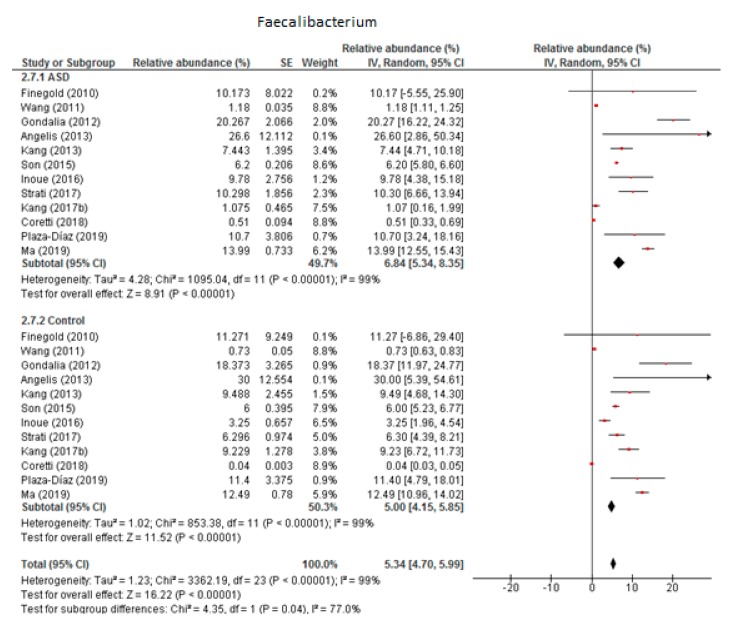
Forest plot of relative abundance of Faecalibacterium in children with ASD and controls.

**Figure 5 nutrients-12-00792-f005:**
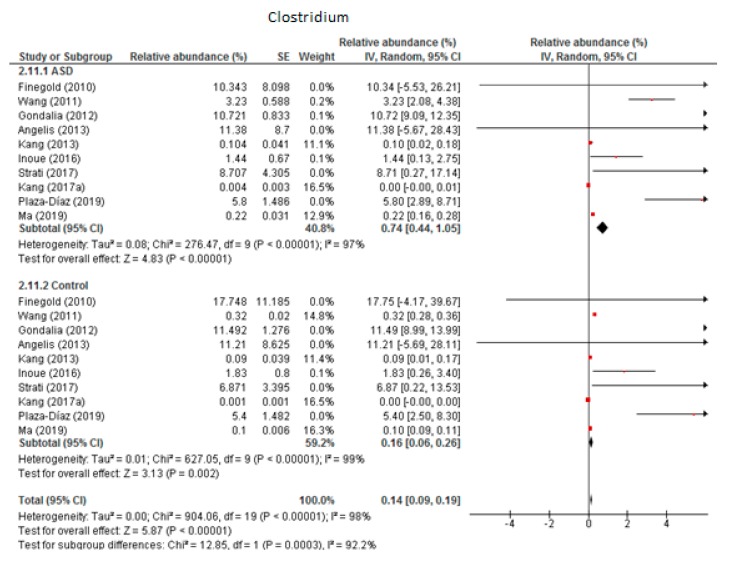
Forest plot of relative abundance of Clostridium in children with ASD and controls.

**Figure 6 nutrients-12-00792-f006:**
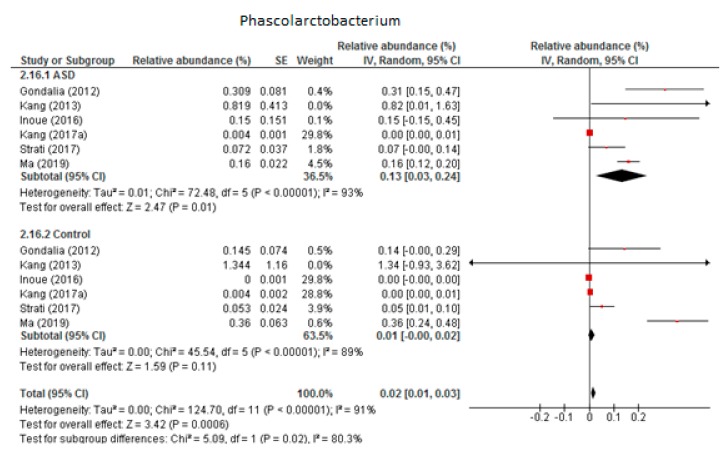
Forest plot of relative abundance of Phascolarctobacterium in children with ASD and controls.

**Figure 7 nutrients-12-00792-f007:**
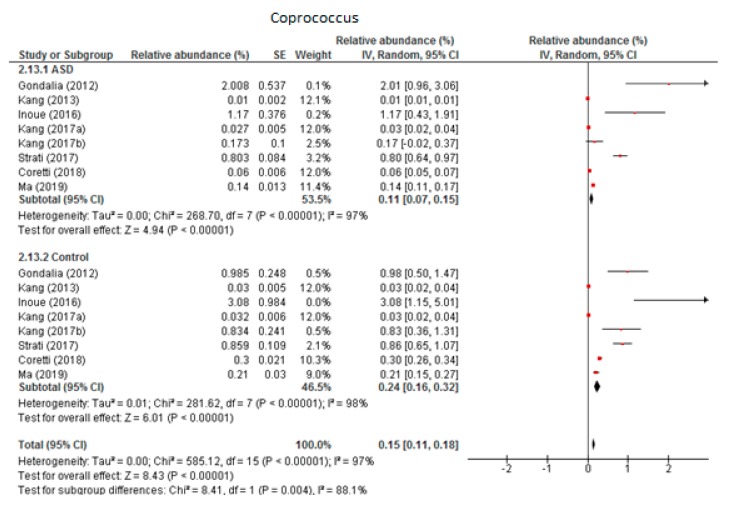
Forest plot of relative abundance of Coprococcus in children with ASD and controls.

**Figure 8 nutrients-12-00792-f008:**
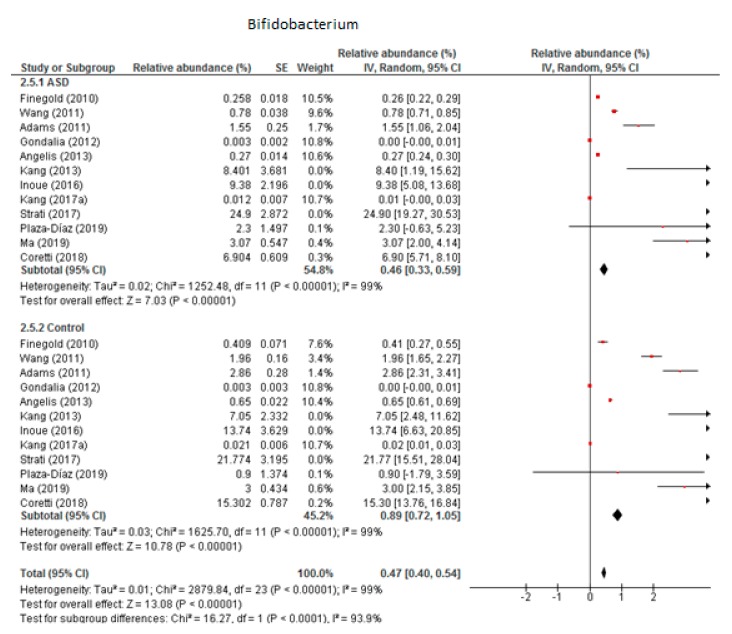
Forest plot of relative abundance of Bifidobacterium in children with ASD and controls.

**Table 1 nutrients-12-00792-t001:** Characteristics of the studies included in the meta-analysis.

Study	Country	ASD (n)	Age (years)	Control (n)	Age (years)	Bacteria Detected	Microbiology Assessment	Dietary Assessment	Probiotics Usage Assessment
Finegold et al., 2010 [31]	USA	11	2–13	8	2–13	**Phylum:** Bacteroidetes, Firmicutes, Proteobacteria, Actinobacteria, Cyanobacteria, Fusobacteria, Verrucomicrobia, Tenericutes **Genus:** *Akkermansia*, *Bacteroides*, *Bifidobacterium*, *Clostridium*, *Faecalibacterium*, *Parabacteroides*, *Ruminococcus*	Pyrosequencing	-	-
Wang et al., 2011 [35]	Australia	23	10.25 ± 0.75	9	9.5 ± 1.25	**Genus:***Akkermansia*, *Bacteroides*, *Bifidobacterium*, *Faecalibacterium*, *Clostridium*	PCR	Only in ASD	Only in ASD
Adams et al., 2011 [11]	USA	58	6.91 ± 3.4	39	7.7 ± 4.4	**Genus:** *Bifidobacterium*	Culture	-	Yes
Gondalia et al., 2012 [20]	Australia	28	2–12	25	2–12	**Phylum:** Bacteroidetes, Firmicutes, Proteobacteria, Actinobacteria, Cyanobacteria, Fusobacteria, Verrucomicrobia, Tenericutes **Genus:** *Anaerostipes*, *Anaerotruncus*, *Bacteroides*, *Bifidobacterium*, *Blautia*, *Clostridium*, *Faecalibacterium*, *Parabacteroides*, *Ruminococcus*, *Sutterella*, *Veillonella*, *Coprococcus*, *Dialister*, *Dorea*, *Roseburia*, *Phascolarctobacterium*	Culture	-	Yes
Angelis et al., 2013 [36]	Italy	10	4–10	10	4–10	**Genus:***Akkermansia*, *Bacteroides*, *Bifidobacterium*, *Clostridium*, *Faecalibacterium*, *Parabacteroides*, *Ruminococcus*	Pyrosequencing	-	No
Kang et al., 2013 [32]	USA	20	6.7 ± 2.7	20	8.3 ± 4.4	**Phylum:** Bacteroidetes, Firmicutes, Proteobacteria, Actinobacteria, Cyanobacteria, Fusobacteria, Verrucomicrobia, Tenericutes. **Genus:** *Akkermansia*, *Anaerostipes*, *Anaerotruncus*, *Bacteroides*, *Bifidobacterium*, *Blautia*, *Clostridium*, *Faecalibacterium*, *Parabacteroides*, *Ruminococcus*, *Sutterella*, *Veillonella*, *Coprococcus*, *Dialister*, *Dorea*, *Phascolarctobacterium*, *Roseburia*	Pyrosequencing	Yes	Yes
Wang et al., 2013 [34]	Australia	23	10.25 ± 0.75	9	9.5 ± 1.25	**Genus:***Ruminococcus*, *Sutterella*	PCR	-	-
Son et al., 2015 [21]	USA	34	7–14	31	7–14	**Phylum:** Bacteroidetes, Firmicutes, Proteobacteria, Actinobacteria, Cyanobacteria, Tenericutes **Genus:** *Faecalibacterium, Sutterella*	PCR	Yes	No
Inoue et al., 2016 [40]	Japan	6	3–5	6	3–5	**Genus:** *Akkermansia, Anaerostipes, Anaerotruncus, Bacteroides, Bifidobacterium, Blautia, Clostridium, Faecalibacterium, Parabacteroides, Ruminococcus, Sutterella, Veillonella, Coprococcus, Dialister, Dorea, Phascolarctobacterium, Roseburia*	Pyrosequencing	-	-
Strati et al., 2017 [37]	Italy	40	5–17	40	5–17	**Phylum:** Bacteroidetes, Firmicutes, Proteobacteria, Actinobacteria, Cyanobacteria, Fusobacteria, Verrucomicrobia **Genus:** *Akkermansia, Anaerostipes, Anaerotruncus, Bacteroides, Bifidobacterium, Blautia, Clostridium, Faecalibacterium, Parabacteroides, Ruminococcus, Sutterella, Veillonella, Coprococcus, Dialister, Dorea, Phascolarctobacterium, Roseburia*	Pyrosequencing	-	No
Kang et al., 2017(a) [12]	USA	18	7–16	20	7–16	**Phylum:** Bacteroidetes, Firmicutes, Proteobacteria, Actinobacteria, Cyanobacteria, Fusobacteria, Verrucomicrobia, Tenericutes **Genus:** *Anaerostipes, Bacteroides, Bifidobacterium, Blautia, Clostridium, Parabacteroides, Ruminococcus, Sutterella, Coprococcus, Dialister, Dorea, Phascolarctobacterium, Roseburia*	PCR	Yes	No
Kang et al., 2017(b) [33]	USA	23	4–17	21	4–17	**Genus:** *Akkermansia, Anaerostipes, Bacteroides, Faecalibacterium, Coprococcus, Roseburia*	Pyrosequencing	Yes	-
Coretti et al., 2018 [17]	Italy	11	2–4	14	2–4	**Phylum:** Bacteroidetes, Firmicutes, Proteobacteria, Actinobacteria **Genus:** *Bacteroides, Bifidobacterium, Blautia, Faecalibacterium, Parabacteroides, Ruminococcus, Coprococcus, Roseburia*	PCR	Yes	-
Pulikkan et al., 2018 [18]	India	30	3–16	24	3–16	**Phylum:** Bacteroidetes, Firmicutes, Proteobacteria, Actinobacteria, Tenericutes	PCR	-	-
Zhang et al., 2018 [38]	China	35	4.9 ± 1.5	6	4.6 ± 1.1	**Phylum:** Bacteroidetes	PCR	No	No
Ma et al., 2019 [16]	China	45	6–9	45	6–9	**Phylum:** Bacteroidetes, Firmicutes, Proteobacteria, Actinobacteria, Cyanobacteria, Fusobacteria, Verrucomicrobia, Tenericutes **Genus:** *Bacteroides, Bifidobacterium, Blautia, Clostridium, Faecalibacterium, Parabacteroides, Ruminococcus, Coprococcus, Phascolarctobacterium, Roseburia*	PCR	Yes	No
Plaza-Díaz et al., 2019 [41]	Spain	48	2–6	57	2–6	**Phylum:** Bacteroidetes, Firmicutes, Proteobacteria, Actinobacteria, Verrucomicrobia **Genus:** *Akkermansia, Bacteroides, Bifidobacterium, Clostridium, Faecalibacterium, Parabacteroides, Ruminococcus, Veillonella*	PCR	Yes	-
Liu et al., 2019 [39]	China	30	2.5–18	20	2.5–18	**Phylum:** Bacteroidetes, Firmicutes, Proteobacteria, Actinobacteria, Cyanobacteria, Fusobacteria, Verrucomicrobia, Tenericutes	PCR	Yes	No

**Table 2 nutrients-12-00792-t002:** Results of the meta-analysis at phylum and genus levels in ASD children and controls, and the significance of the difference between them.

	Studies Included	ASD			Control			Overall Effect	Subgroup Differences	
		Overall Relative Abundance	95% CI	Between-Study I^2^	Overall Relative Abundance	95% CI	Between-Study I^2^		I^2^	*P*
Bacteroidetes	12	14.33	12.79–15.87	100	10.97	9.57–12.36	100	25.57	90	**0.002**
*Bacteriodes*	12	9.04	7.62–10.47	99	4.69	3.67–5.71	99	15.56	95.8	**<0.001**
*Parabacteroides*	10	0.32	0.21–0.43	95	0.04	−0.00–0.08	92	5.69	95.3	**<0.001**
**Firmicutes**	11	13.42	12.50–14.34	100	10.77	9.89–11.64	100	25.82	94	**<0.001**
*Anaerostipes*	6	0.01	−0.01–0.03	83	0.02	0.00–0.04	92	2.67	0	0.620
*Anaerotruncus*	4	0.23	0.05–0.41	89	0.14	0.00–0.27	89	2.66	0	0.430
*Blautia*	6	1.52	0.86–2.19	98	1.91	0.61–3.21	99	11.92	0	0.610
*Faecalibacterium*	12	6.84	5.34–8.35	99	5.00	4.15–8.35	99	16.22	77	0.040
*Ruminococcus*	11	2.90	2.22–3.58	96	2.21	1.50–2.92	98	11.41	47.3	0.170
*Veillonella*	5	0.07	0.03–0.11	20	0.13	−0.04–0.31	75	2.64	0	0.460
*Clostridium*	10	0.74	0.44–1.05	97	0.16	0.06–0.26	98	5.87	92.2	**<0.001**
*Coprococcus*	8	0.11	0.07–0.15	97	0.24	0.16–0.32	98	8.43	88.1	**0.004**
*Dialister*	5	0.54	0.09–0.99	84	0.65	0.09–1.22	81	2.59	0	0.760
*Dorea*	5	0.42	0.18–0.66	97	0.21	0.09–0.32	94	3.79	60.4	0.110
*Phascolarctobacterium*	6	0.13	0.03–0.24	93	0.01	−0.00–0.02	89	3.42	80.3	**0.020**
*Roseburia*	7	0.11	0.04–0.19	92	0.09	0.02–0.15	94	4.54	0	0.630
**Proteobacteria**	11	0.09	0.05–0.13	97	0.02	0.00–0.03	96	4.2	92.6	**<0.001**
*Sutterella*	7	0.11	0.06–0.17	99	0.22	−0.06–0.50	100	4.84	0	0.480
**Actinobacteria**	11	0.53	0.38–0.69	97	0.43	0.29–0.58	97	7.86	0	0.360
*Bifidobacterium*	12	0.46	0.33–0.59	99	0.89	0.72–1.05	99	13.08	93.9	**<0.001**
**Cyanobacteria**	7	0.00	0.00–0.01	70	0.01	0.00–0.01	84	2.66	0	0.440
**Fusobacteria**	7	0.02	0.00–0.03	97	0.04	0.01–0.08	100	3.31	0	0.430
**Verrucomicrobia**	8	0.04	0.01–0.09	88	0.07	0.01–0.14	85	3.19	0	0.430
*Akkermansia*	8	0.04	−0.10–0.18	81	0.55	−0.36–1.46	49	1.25	14.4	0.280
**Tenericutes**	7	0.06	0.04–0.07	99	0.00	0.00–0.00	28	5.38	97.7	**<0.001**

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
