# Peer review of "Composition of Gut Microbiota in Children with Autism Spectrum Disorder: A Systematic Review and Meta-Analysis"

_nutrients, 2020, doi:10.3390/nu12030792_

Round 1

Reviewer 1 Report

This is an excellent and thoughtful metanalysis of existing studies on the gut microbiome and ASD, a very important area in brain-gut research.

I thought the analysis methods were well descibed and justified.

I especially liked the discussion and the conservative but valuable way the authors put the data in perspective without "overinterpretation" of the data.

Excellent job!

Author Response

We really appreciate your words about the present work. Thank you very much.

Reviewer 2 Report

Vázquez et al performed the meta-analysis of microbiome composition on the published references and aim to describe and compare the composition of gut microbiota in children with and without ASD. The criteria for the selection of publications subjected to meta-analysis is very stringient and careful. The writing is clear and detailed. The authors have a very clear rationale for the initiation of this systemic review. This systemic review is timely and needed for the ASD and microbiome field. It will help the scientists to narrow down the bacterial targets for ASD therapy. Following below are the comments I have after reading this comprehensive review.

It would be of interest to provide the meta-analysis of adult ASD (>18yrs). I understand the rationale to exclude adult data. On the contrary, microbiota will be more stabilized at the adult stage based on studies. Many environmental factors would shape the development of gut microbiota during the childhood stage or adolescent stage. Perhaps meta-analysis on adult cohorts would provide information that is more stabilized. Plus, searching for the microbiota difference in adulthood could also provide direction for probiotic therapy on adult ASD.

Are there any controversial data in any of the phyla or genera in these 18 studies? The discussion should list those data that are contradicted.

There are totally 9 out of 18 studies included diet assessment. Would it be possible to run an analysis on the association between diet and microbiota? Same issue with the three probiotics assessment studies. It would be of interest to learn more correlations between the diet, probiotics, and microbiota in ASD populations.

Are there any symptoms closely associated with the microbiome meta-analysis? The ADOS or ADI-R could provide in-depth analysis.

The ratio of Bacteroidetes/Firmicutes is shifted in many diseases/disorders. Since both phyla of Bacteroidetes and Firmicutes are more abundant in the ASD cohorts. It would be of interest to consider putting the ratio of Bacteroidetes/Firmicutes into the meta-analysis.

Are there findings in species-level? The information of phyla and genera are important. However, what the field urged to know will be the differences in species level.

A paragraph of "Bacterial phyla Less Abundant in ASD Children" is missing in the result section.

Figures (1-8) are missing in the entire manuscript.

Line 32, Please provide a reference for “a neurobiological etiology that developed during the first years of life”
